



SOIL

# A new look at an old concept: using $^{15}N_2O$ isotopomers to understand the relationship between soil moisture and $N_2O$ production pathways

**Katelyn A. Congreves[1], Trang Phan[1], and Richard E. Farrell[2]**

[1]Department of Plant Sciences, University of Saskatchewan, Saskatoon, SK S7N 5A8, Canada
[2]Department of Soil Science, University of Saskatchewan, Saskatoon, SK S7N 5A8, Canada

**Correspondence:** Kate A. Congreves (kate.congreves@usask.ca)

**Abstract.** Understanding the production pathways of potent greenhouse gases, such as nitrous oxide ($N_2O$), is essential for accurate flux prediction and for developing effective adaptation and mitigation strategies in response to climate change. Yet there remain surprising gaps in our understanding and precise quantification of the underlying production pathways – such as the relationship between soil moisture and $N_2O$ production pathways. A powerful, but arguably underutilized, approach for quantifying the relative contribution of nitrification and denitrification to $N_2O$ production involves determining $^{15}N_2O$ isotopomers and $^{15}N$ site preference (SP) via spectroscopic techniques. Using one such technique, we conducted a short-term incubation where $N_2O$ production and $^{15}N_2O$ isotopomers were measured 24 h after soil moisture treatments of 40 % to 105 % water-filled pore space (WFPS) were established for each of three soils that differed in nutrient levels, organic matter, and texture. Relatively low $N_2O$ fluxes and high SP values indicted nitrification during dry soil conditions, whereas at higher soil moisture, peak $N_2O$ emissions coincided with a sharp decline in SP, indicating denitrification. This pattern supports the classic $N_2O$ production curves from nitrification and denitrification as inferred by earlier research; however, our isotopomer data enabled the quantification of source partitioning for either pathway. At soil moisture levels < 53 % WFPS, the fraction of $N_2O$ attributed to nitrification ($F_N$) predominated but thereafter decreased rapidly with increasing soil moisture ($x$), according to $F_N = 3.19 - 0.041x$, until a WFPS of 78 % was reached. Simultaneously, from WFPS of 53 % to 78 %, the fraction of $N_2O$ that was attributed to denitrification ($F_D$) was modelled as $F_D = -2.19 + 0.041x$; at moisture levels of > 78 %, denitrification completely dominated. Clearly, the soil moisture level during transition is a key regulator of $N_2O$ production pathways. The presented equations may be helpful for other researchers in estimating $N_2O$ source partitioning when soil moisture falls within the transition from nitrification to denitrification.

## 1 Introduction

Agricultural soils are the largest source of anthropogenic $N_2O$ emissions, accounting for up to 80 % TS1 of total $N_2O$ emissions (Environment Canada, 2019). Understanding the mechanisms leading to the emission of this potent greenhouse gas is essential for accurate flux prediction and for developing effective adaptation and mitigation strategies in response to climate change. Decades of research have strengthened our understanding of $N_2O$ fluxes – namely, how $N_2O$

production is regulated by soil oxygen, substrate availability, and microbial activity (Butterbach-Bahl et al., 2013; Chapuis-Lardy et al., 2007; Wagner-Riddle et al., 2017) as well as how $N_2O$ emission is regulated by advection, solubility, and diffusion (Balaine et al., 2013; Clough et al., 2005). Indeed, our understanding of the relationship between $N_2O$ production and soil moisture has benefited greatly from the use of $^{15}N$ tracers (Bateman and Baggs, 2005; Stevens and Laughlin, 1997; Groffman et al., 2006). However, there remain surprising grey areas in our understanding of the un-

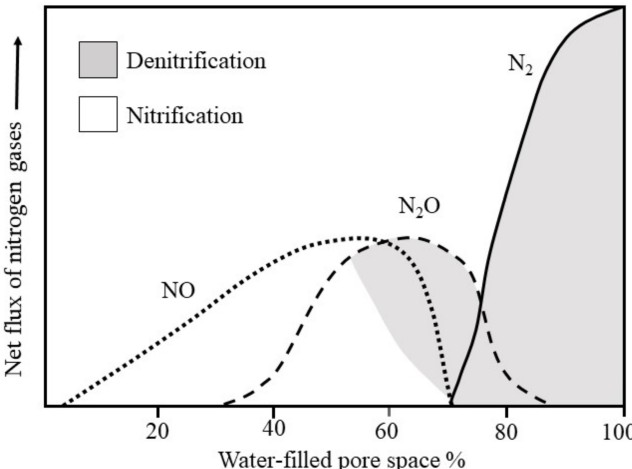

**Figure 1.** Relative contributions of nitrification and denitrification processes to $N_2O$ production as a function of water-filled pore space (adapted from Davidson, 1991).

derlying mechanisms, with one such area being the precise relationship between soil moisture and $N_2O$ production pathways, especially during the transition from one dominant pathway to another (Bateman and Baggs, 2005).

Nitrous oxide is a product of nitrification and denitrification – microbially driven processes that depend on the aeration status of the soil (Banerjee et al., 2016; Barnard et al., 2005). As a result, the relative contributions of nitrification and denitrification are often determined based on their relationship to soil water-filled pore space (WFPS), which acts as a proxy for aeration status. However, the widely cited relationship between soil $N_2O$ production and soil moisture (Fig. 1) is actually an educated deduction that blends work from two different studies, from which the $N_2O$ production pathways are inferred (Davidson, 1991; Linn and Doran, 1984). As such, it may be argued that the precise relationship between soil water content and $N_2O$ production mechanisms remains unclear and requires more complete quantification. While previous research has provided important steps towards better quantifying the relationship using [15]N enrichment and acetylene inhibition techniques (Bateman and Baggs, 2005), natural abundance [15]N techniques may provide superior information by imposing fewer confounding effects on stepwise N transformations.

Isotopomers – i.e., isomers having the same number of each isotopic atom but differing in their position (IUPAC, 1997; Ostrom and Ostrom, 2012) – provide a powerful and novel approach for quantifying the relative contribution of $N_2O$ producing processes via nitrification and denitrification (Van Groenigen et al., 2015). Early work focused on the intramolecular distribution of [15]N within the linear $N_2O$ molecule (Sutka et al., 2006; Toyoda et al., 2005), investigations of atmospheric or oceanic $N_2O$ isotopomers (Popp et al., 2002; Toyoda and Yoshida, 1999; Yoshida and Toy-

oda, 2000), and soil-emitted $N_2O$ isotopomers (Perez et al., 2001; Yamulki et al., 2001). The isotopomers of $N_2O$ (i.e., [14]N[15]NO and [15]N[14]NO) can be quantified using advanced laser spectroscopic approaches – including cavity ring-down spectroscopy (CRDS) – that enable the intramolecular [15]N distribution of $N_2O$ to be determined (Mohn et al., 2014). The difference between the abundance of [15]N within the central (alpha – $\alpha$) and the terminal (beta – $\beta$) N atoms of the linear $N_2O$ molecule is expressed as site preference (SP), and high SP values of 13 to 37 ‰ are attributed to nitrification (hydroxylamine oxidation), while SP values of 0 ‰ or less indicate nitrite or nitrate reduction (denitrification and nitrifier denitrification; Denk et al., 2017; Ostrom et al., 2010; Sutka et al., 2006; Toyoda et al., 2005). The underlying reason for the distinct differences in SP values of $N_2O$ from either microbial pathway is due to primary kinetic isotope effects when $N_2O$ is produced (Popp et al., 2002).

Our objective was to use [15]$N_2O$ isotopomers to precisely quantify the relationship between soil moisture and $N_2O$ production in soils differing in soil nutrient level, organic matter, and texture.

## 2   Materials and methods

### 2.1   Soil collection and characterization

Surface (0–10 cm) soils representing different nutrient levels and texture classes were collected from three locations in the Dark Brown soil zone in Saskatchewan, Canada. The soils – classified as Dark Brown chernozems of the Sutherland, Asquith, and Bradwell associations – were collected using a shovel and air dried, and sub-samples were shipped to A&L Laboratories Inc. (London, ON) for analysis (Table 1). For additional characterization, sub-samples were analyzed at the University of Saskatchewan for equilibrium soil water content, soil inorganic N levels, soil total N concentration, and [15]N abundance (Table 1). The equilibrium soil water was determined via the long-column method based on the average of four technical replicates (Reynolds and Topp, 2007). Initial soil $NO_3^-$ and $NH_4^+$ concentrations were determined in quadruplicate using the KCl extraction method of Maynard et al. (2007); briefly, 5 g of soil was mixed with 50 mL of 2 M KCl, shaken for 30 min, and filtered through Whatman 42 filter paper; the extracts were frozen at $-20\,^{\circ}C$ until they could be analyzed. For analysis, the extracts were thawed and allowed to equilibrate to room temperature before being analyzed using air-segmented (continuous) flow analysis with a SEAL AA3 HR chemistry analyzer (SEAL Analytical, Kitchener, ON). Soil total N concentration (%) and [15]N content (atom %) were determined in duplicate using a Costech ECS 4010 elemental analyzer (Costech Analytical Technologies Inc., Valencia, CA) coupled to a high-precision Delta V mass spectrometer (Bremen, Germany) with a precision of 0.06 ‰ for $\delta^{15}N$. Chickpea flour with an atom % [15]N = 0.3691 was used as a lab reference.

**Table 1.** Soil physical and chemical characteristics.

|  | Sutherland | Asquith | Bradwell |
| --- | --- | --- | --- |
| Previous cropping history | Vegetable crops | Fodder crops | Field crops |
| Texture class | Silty clay loam | Sandy loam | Loam |
| Organic matter (%) | 5.9 | 3.9 | 2.7 |
| Equilibrium soil water ($\theta_g$) | 0.46 | 0.40 | 0.33 |
| pH | 7.6 | 7.5 | 7.9 |
| Cation exchange capacity (CEC; $\text{cmol}_c\,\text{kg}^{-1}$) | 34.8 | 18.6 | 16.9 |
| Total N (%) | 0.42 | 0.21 | 0.16 |
| Total $^{15}$N (atom %) | 0.371 | 0.370 | 0.368 |
| Nitrate ($\mu\text{g}\,\text{g}^{-1}$) | 194 | 35 | 10 |
| Ammonium ($\mu\text{g}\,\text{g}^{-1}$) | 3.8 | 1.7 | 5.2 |
| Bray phosphorus (ppm) | 542 | 190 | 23 |
| Potassium (ppm) | 1415 | 544 | 329 |
| Sulfur (ppm) | 49 | 28 | 13 |
| Magnesium (ppm) | 925 | 448 | 432 |
| Calcium (ppm) | 4650 | 2670 | 2490 |

## 2.2 Incubation experimental design

For the incubation study, soil microcosms were established over a range of moisture treatments for each soil and arranged in a completely randomized design with four replicates. For each microcosm, sieved (2 mm mesh screen) and air-dried soil was packed into a small (5.9 cm inner diameter, 0.80 cm tall) plastic petri dish. The mass of soil needed to fill the petri dish varied with texture – ranging from 22.0 to 29.0 g – and yielded soil bulk densities of 1.01, 1.10, and 1.33 g cm$^{-3}$ for the Sutherland, Asquith, and Bradwell soils, respectively. While the quantities and bulk densities differed for each soil type, it was essential that the soil completely fill the petri dishes to avoid any differences in soil surface boundary layer or gas diffusion that would alter N$_2$O emission.

Soil moisture treatments were based on gravimetric soil water content ($\theta_g$) established by adding deionized water to the soil microcosms, using a fine mist of water applied from a manual spray bottle, to a predetermined weight. Gravimetric soil moisture content was varied to yield a WFPS between 40 % and 105 %.

The gravimetric water, volumetric water ($\theta_v$), and WFPS were determined according to Eqs. (1)–(3):

$$\theta_g \left( \text{g}\,\text{H}_2\text{O}\,\text{g}\,\text{soil}^{-1} \right)$$
$$= \frac{\text{water added (g)}}{\text{dry soil (g)}}, \tag{1}$$

$$\theta_v \left( \text{cm}^3\,\text{H}_2\text{O}\,\text{cm}^3\,\text{soil}^{-1} = \theta_{ag} \times \text{BD} \right), \tag{2}$$

$$\%\text{WFPS} = \left[ \frac{\theta_v}{\left( 1 - \frac{\text{BD}}{\text{PD}} \right)} \right] \times 100, \tag{3}$$

where BD denotes soil bulk density and PD denotes particle density (PD), which was assumed to be 2.65 g cm$^{-3}$.

Immediately after moistening the soil microcosm, the petri dish was sealed inside a 1 L wide-mouth mason jar fitted with a gas sampling septum, and the time of sealing was recorded. Blank jars containing an empty petri dish were set up to account for background (atmospheric) gas concentrations. The microcosms were incubated at 22 °C ± 1 °C for 24 h.

## 2.3 Sampling and analysis

After 24 h, a headspace gas sample was collected from each microcosm (with the time of sampling recorded) using a 20 mL plastic syringe fitted with a 22-gauge needle, injected into an evacuated 12 mL Exetainer® tube (Labco Limited, UK), and analyzed for N$_2$O, CO$_2$, and O$_2$ concentration using gas chromatography (Bruker 450 GC, Bruker Biosciences, Billerica, MA). Immediately thereafter, a separate 30 mL gas sample was collected from each microcosm, injected into an evacuated 12 mL Exetainer® tube, and analyzed for $^{15}$N$_2$O concentration, $\delta^{15}\text{N}_\alpha$, $\delta^{15}\text{N}_\beta$, and $\delta^{18}$O using a CRDS-based Picarro G5131-*i* isotopic N$_2$O analyzer (Picarro Inc., Santa Clara, CA).

## 2.4 Isotopomer approach using $^{15}$N site preference and $\delta^{18}$O for N$_2$O source identification

Site preference was calculated by subtracting the abundance of $^{15}$N from the terminal N atom (beta – $\beta$) from that of the central (alpha – $\alpha$) N atom. The fraction of N$_2$O derived from hydroxylamine oxidation during nitrification ($F_N$) or the reduction of nitrate or nitrite during denitrification ($F_D$) was estimated for each soil by adopting the isotopomer mixing approach used by others (Deppe et al., 2017; Lewicka-Szczebak et al., 2017; Zou et al., 2014) and which use the $^{15}$N SP and $\delta^{18}$O values of gas samples collected from the soils. As suggested by Lewicka-Szczebak et al. (2017) and

by Well et al. (2012), and because SP was more closely correlated to $\delta^{18}$O ($r = 0.906$) than $\delta^{15}$N ($r = 0.849$), we used the relationship between $\delta^{15}$N SP and $\delta^{18}$O instead of $\delta^{15}$N SP and bulk $\delta^{15}$N. Equations (4) and (5) show the source partitioning calculations:

$$F_{\rm N} = \frac{{\rm SP}_x - {\rm SP}_{\rm D}}{{\rm SP}_{\rm N} - {\rm SP}_{\rm D}}, \tag{4}$$

$$F_{\rm D} = 1 - F_{\rm N}, \tag{5}$$

where $F_{\rm N}$ and $F_{\rm D}$ indicate the fraction of $N_2O$ derived from nitrification or denitrification, respectively; SP denotes the site preference for the sample (${\rm SP}_x$) and the endmembers for nitrification (${\rm SP}_{\rm N}$) and denitrification (${\rm SP}_{\rm D}$).

Rather than relying on average literature-derived endmembers like in previous work (Deppe et al., 2017; Lewicka-Szczebak et al., 2017), we used soil-specific endmembers derived from our data to perform the linear mixed model. This is because we measured a wide range of soil WFPS treatments with high frequency between dry and moist conditions for each soil, enabling us to determine the point at which the $\delta^{15}$N SP or $\delta^{18}$O values either dropped or increased as soil WFPS changed (as precisely as the data permitted), indicating a transition from nitrification to denitrification. This approach is consistent with earlier recommendations that data be collected at high enough frequencies to capture gradual changes in isotope values as influenced by traditional proxies (i.e., gradual changes in soil WFPS; Decock and Six, 2013a). However, it must be noted that the underlying assumption is that the soil-specific endmembers are more reflective of the transition from nitrification to denitrification in each of the soils tested herein than general literature-derived endmembers would be for any one soil. Moreover, it is assumed that the endmembers represent $N_2O$ produced when the sole source was either nitrification or denitrification. Endmembers for ${\rm SP}_{\rm D}$ to ${\rm SP}_{\rm N}$ were set at 2.0 to 23.7, 0.7 to 21.7, and 14.4 to 23.3 for the Sutherland, Asquith, and Bradwell soils, respectively. Endmembers for $\delta^{18}O_{\rm D}$ to $\delta^{18}O_{\rm N}$ were set at 16.0 to 35.1, 18.8 to 39.5, and 25.4 to 34.2 for the Sutherland, Asquith, and Bradwell soils, respectively. The endmember ranges were based on our data, where ${\rm SP}_{\rm N} / \delta^{18}O_{\rm N}$ represented the average values before the transition zone from nitrification- to denitrification-dominated $N_2O$ production; ${\rm SP}_{\rm D} / \delta^{18}O_{\rm D}$ represented the lowest values during denitrification for each soil type. For source partitioning, the influence of $N_2O$ reduction to $N_2$ on SP was taken into account by using the reduction and mixing line intercept approach – as described by Deppe et al. (2017) and Lewicka-Szczebak et al. (2017). However, rather than using an estimated reduction line derived from the literature, we calculated the slope and intercept for the reduction line based on our data: the SP / $\delta^{18}$O plot for the soil moisture range after the transition zone for each soil type. The reduction line was placed through the average SP value of gas samples derived from the < 60 % water-filled pore space range for each soil. The point

of intersection between the endmember mixing line and the reduction line gave the estimated initial isotope values (SP* and $^{18}$O) of produced $N_2O$ before reduction to $N_2$. In the soil moisture range after the transition from nitrification to denitrification, if the SP* value was higher than the measured SP value of the gas sample, the measured SP value was used, since $N_2O$ reduction was assumed to be negligible. The $F_{\rm N}$ and $F_{\rm D}$ were then calculated from SP values (or SP*) and the SP values of the nitrification and denitrification endmembers. This calculation was done for each soil type separately.

## 2.5 Statistical analysis

Correlation and linear regression analyses were conducted in CoStat (CoStat ver. 6.451 (CoHort Software, Monterey, CA)) to determine associations between soil moisture and SP.

## 3 Results and discussion

### 3.1 Nitrous oxide production

Nitrous oxide production during the 24 h incubation varied dramatically among the three soils, with peak $N_2O$ production occurring at soil water contents equivalent to 70 %–80 % WFPS (Fig. 2). Peak $N_2O$ production was 100-fold greater from the Sutherland soil (100 ng $N_2O$-N $g^{-1}$ 24 $h^{-1}$) compared to the Bradwell soil (1 ng $N_2O$-N $g^{-1}$ 24 $h^{-1}$), and about 4-fold greater than that from the Asquith soil (24 ng $N_2O$-N $g^{-1}$ 24 $h^{-1}$; Fig. 2). This differentiation follows the same trend as soil inorganic N availability and soil organic matter, which decreased in the order Sutherland > Asquith > Bradwell (Table 1).

Regardless of the amount of $N_2O$ evolved, there were similarities in how soil moisture influenced relative $N_2O$ production. For all soil types, relatively low $N_2O$ fluxes were associated with drier soil treatments; $N_2O$ fluxes were incrementally magnified as soil moisture levels increased from about 55 % to 80 % WFPS (Fig. 2a, b, and c). At soil moisture levels exceeding $\sim 80$ % WFPS, fluxes either remained relatively high, as was the case for the Sutherland soil, or decreased slightly, as was observed for the Asquith and Bradwell soils.

### 3.2 Nitrous oxide $^{15}$N site preference, $\delta^{15}$N, and $\delta^{18}$O

Not only total $N_2O$ concentration, but the $^{15}$N SP, $\delta^{15}$N, and $\delta^{18}$O of $N_2O$ changed with soil moisture level in parallel with each other (Fig. 2d, e, and f). We identified three moisture ranges – differing slightly for each soil (Table 2) – that regulated $N_2O$ production pathways based on distinct SP, $\delta^{15}$N, and $\delta^{18}$O values (Fig. 2).

For each soil, the $\delta^{15}$N and $\delta^{18}$O values decreased in the same soil moisture region in which the SP values decreased (Fig. 2d, e, and f). Based on the patterns for $N_2O$ fluxes and SP, $\delta^{15}$N, and $\delta^{18}$O values as related to soil moisture (Fig. 2;

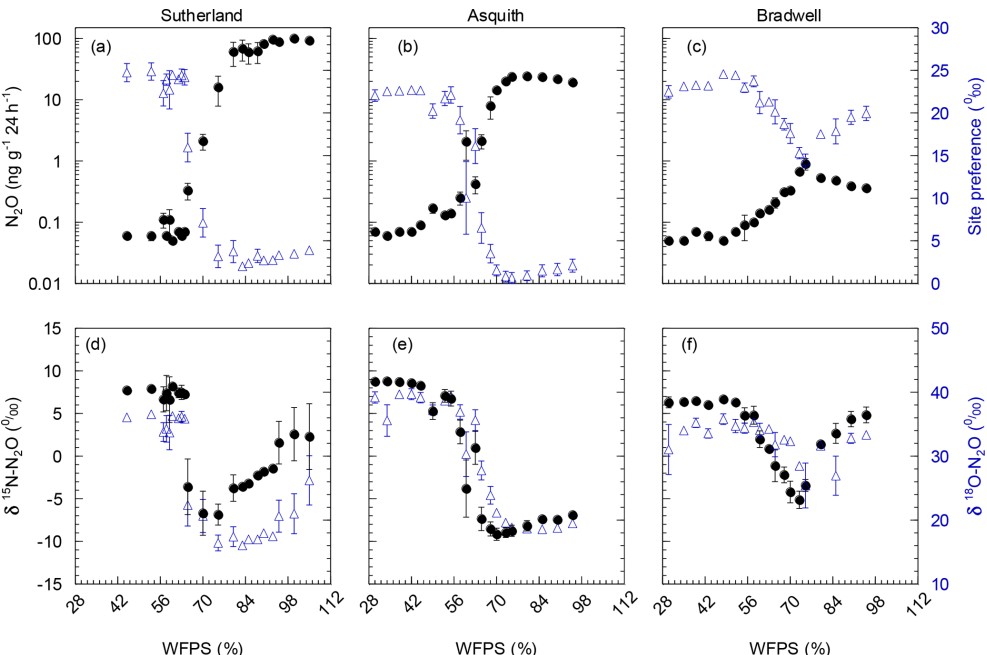

**Figure 2. (a, b, c)** $N_2O$ production as influenced by soil water-filled pore space (WFPS; black; left axis); corresponding $^{15}N_2O$ isotopomer site preference (SP; blue; right axis). **(d, e, f)** $\delta$bulk$^{15}N$ (black; left axis) and $\delta^{18}O$ (blue; right axis) of emitted $N_2O$ as influenced by soil water-filled pore space (WFPS). Note: $N_2O$ emissions were plotted on a $\log_{10}$ scale to accommodate the large range in emissions from the different soils. Data points represent means ($n = 4$), and bars represent the standard errors.

**Table 2.** Linear regressions between $^{15}N$ site preference and soil water-filled pore space (%) during three soil moisture regions for each soil type (i) before the transition from nitrification, (ii) during the transition from nitrification to denitrification, and (iii) after the transition to denitrification. Note: "ns" denotes not significant and * and ** denote significance at $p < 0.05$ and $p < 0.01$, respectively. CE2

| Soil type | WFPS (%) | Slope | Intercept | Pearson $r$ | $p$ value |
|---|---|---|---|---|---|
| Before transition | | | | | |
| Sutherland | < 64 | −0.049 | 26.69 | −0.30 | 0.4660[ns] |
| Asquith | < 58 | −0.004 | 22.04 | −0.04 | 0.8973[ns] |
| Bradwell | < 63 | 0.010 | 22.69 | 0.14 | 0.6781[ns] |
| During transition | | | | | |
| Sutherland | 64–83 | −0.99 | 81.62 | 0.88 | 0.0214* |
| Asquith | 58–73 | −1.19 | 85.75 | −0.89 | 0.0067** |
| Bradwell | 63–75 | −0.59 | 58.29 | −0.99 | 0.0004* |
| After transition | | | | | |
| Sutherland | > 83 | 0.065 | −3.01 | 0.86 | 0.0126* |
| Asquith | > 73 | 0.072 | −4.77 | 0.99 | 0.0064** |
| Bradwell | > 75 | 0.262 | −4.47 | 0.94 | 0.0154* |

Table 2), our results visually indicate that there was a transition between nitrification-derived and denitrification-derived $N_2O$ production at between 64 % and 83 %, 58 % and 75 %, and 63 % and 75 % WFPS for the Sutherland, Asquith, and Bradwell soils, respectively.

Prior to the transition in $N_2O$ production pathway, when the soil was relatively dry, the SP values averaged 23.7‰,

23.3‰, and 21.7‰ from the Sutherland, Asquith, and Bradwell soils, respectively. These values are in line with expected SP values attributed to nitrification (Denk et al., 2017; Ostrom et al., 2010; Sutka et al., 2006; Toyoda et al., 2005). Furthermore, the observed consistency among soil types – and the negligible (near zero) slopes between WFPS and $^{15}N$ SP – suggests that average SPs during nitrifica-

tion are relatively insensitive to the rate of production or associated $N_2O$ accumulation. It is known that isotopic fractionation governed by kinetic isotope effects occurs during the reaction sequence $NH_4^+ \rightarrow NH_2OH \rightarrow NOH \rightarrow NO \rightarrow N_2O$ and $NH_4^+ \rightarrow NO_2 \rightarrow NO \rightarrow N_2O$; however, oxidation of NOH does not involve a primary kinetic isotope effect and thus should not markedly affect SP (Popp et al., 2002).

During the transition from nitrification to denitrification, SP declined rapidly in all soils (Fig. 2; Table 2). The lowest SP values were 2.0 ‰, 0.7 ‰, and 14.4 ‰ for the Sutherland, Asquith, and Bradwell soils, respectively. In general, sharp slopes characterized the decline in SP values with increasing soil moisture during the transition; but the Sutherland and Asquith soils had steeper slopes than the Bradwell soil (Table 2). This difference was likely related to differences in soil inorganic or mineralizable N availability (Table 1) and possibly to differences in the rates of denitrification.

After the transition to denitrification, the SP values increased slightly as soil moisture increased (Table 2) – but more so for the Bradwell soil than for the Sutherland and Asquith soils. This finding supports the sensitivity of SP values to the degree of stepwise completion of denitrification ($N_2O$ reduction to $N_2$). We hypothesize that the ratio of $N_2O$ produced to the $N_2O$ reduced was lowest for the Bradwell soil. Contrary to the large accrual of $N_2O$ from the Sutherland and Asquith soils, the low concentration of $N_2O$ produced from the Bradwell soil likely favoured complete reduction (i.e., tighter "hole in the pipe") – causing the Bradwell soil SP values to be the most sensitive to reduction of $N_2O$ after the transition to denitrification (Fig. 2; Table 2). Correspondingly, using the mapping-model approach to calculate the fraction of denitrified $N_2O$ reduced to $N_2$ (Lewicka-Szczebak et al., 2017), we estimated that much larger fractions of $N_2O$ were reduced to $N_2$ at 95 % WFPS in the Bradwell soil (0.47) compared to the Sutherland or Asquith soils (0.13 to 0.14). The greater amounts of $N_2O$ produced by the nutrient-rich Sutherland and Asquith soils may have overwhelmed any reduction effect on the SP of $N_2O$. Our findings attribute "$N_2O$-leaky" soils to excess inorganic N or mineralization potential.

### 3.3 The hole in the pipe influences site preference

As alluded to above, the Bradwell results were most dissimilar to the other soils. It is intriguing that the SP values for the Bradwell soil $N_2O$ never dropped below 14.4 ‰. While it is clear from the pattern of $N_2O$ fluxes and SP, $\delta^{15}N$, and $\delta^{18}O$ values (Fig. 2) that $N_2O$ production transitioned to denitrification as the soil water content was increased (Table 2), it is curious that the SP values were not lower (i.e., closer to 0 ‰) as earlier work demonstrated for denitrification (Denk et al., 2017; Ostrom et al., 2010; Sutka et al., 2006; Toyoda et al., 2005; Winther et al., 2018). Reasons for this discrepancy are as yet unclear, but we are not alone in finding SP values above 0 ‰ that are attributed to denitrification (Winther

et al., 2018). Differences might be related to differences in microbial community structure and activity, as suggested by Decock and Six (2013a). Also, it is very likely that multiple processes underlying $N_2O$ production and consumption acted simultaneously to cause an SP value that was higher than expected (Decock and Six, 2013a). Otherwise, $N_2O$ reduction to $N_2$ might have played a role that was larger than anticipated for the Bradwell soil. Indeed, SP values within the expected range for bacterial denitrification are known to be sensitive to the reduction of $N_2O$ to $N_2$ (Deppe et al., 2017; Jinuntuya-Nortman et al., 2008; Lewicka-Szczebak et al., 2014; Ostrom et al., 2007; Well and Flessa, 2009). Despite similarities among soils in the robust patterns of how SP values are influenced by soil moisture (Fig. 2; Table 2), SP exhibited a significant ($p<0.0001$) soil by moisture region interaction. This finding agrees with earlier suggestions that, at finer scales, the $^{15}N_2O$ isotopic signatures and SP values are likely regulated by the active soil microbial community, process rates, and soil heterogeneity (Decock and Six, 2013a; Lewicka-Szczebak et al., 2014). Denitrification results in cleavage of the covalent bond between the central N and O in $N_2O$, and based on kinetic isotope fractionation, results in an increase in the $^{15}N$ content of the $\alpha$ position of the residual $N_2O$, thereby increasing the SP (Popp et al., 2002; Ostrom et al., 2007). Thus, the increase in SP in response to $N_2O$ reduction results in a small (but important) shift away from the SP values associated with the origins of denitrification ($\sim 0$ ‰) towards those of nitrification, i.e., 33 ‰ (Sutka et al., 2006). Previously, the fractionation of SP due to $N_2O$ reduction was constrained to a variation of $-2$ ‰ to $-8$ ‰ (Jinuntuya-Nortman et al., 2008; Lewicka-Szczebak et al., 2014; Well and Flessa, 2009). Ostrom et al. (2007) showed that the rate of reduction must be substantially greater than 10 % of that of production to impact the SP estimates of $N_2O$ from denitrification by more than a few percent. Because it is likely that $N_2O$ consumption was greater than production for the Bradwell soil when soil moisture exceeded 75 % WFPS, our results indicate that the size of the hole in the pipe may influence denitrification SP to a greater extent than previously documented.

For $N_2O$ source identification, we adopted an isotopomer mixing approach (Deppe et al., 2017; Lewicka-Szczebak et al., 2017; Zou et al., 2014) and constructed isotopomer maps (i.e., plots of SP vs.. $\delta^{18}O$). This approach allowed us to estimate the impact of $N_2O$ reduction to $N_2$ on SP. Reduction slopes for our three soils averaged at 0.28, which is similar to the literature-derived average of 0.35 or 0.33 used by Deppe et al. (2017) and Lewicka-Szczebak et al. (2017), respectively, though they varied over a wide range, i.e., from 0.16 to 0.52 (Fig. 3). A high reduction slope, such as that observed for the Bradwell soil, might be associated with the magnitude of $N_2O$ production relative to potential nitrous oxide reductase activity or conditions that favour more complete stepwise reduction of $N_2O$ to $N_2$. Whereas the reduction effect on SP might be stronger than previously thought, it may only be

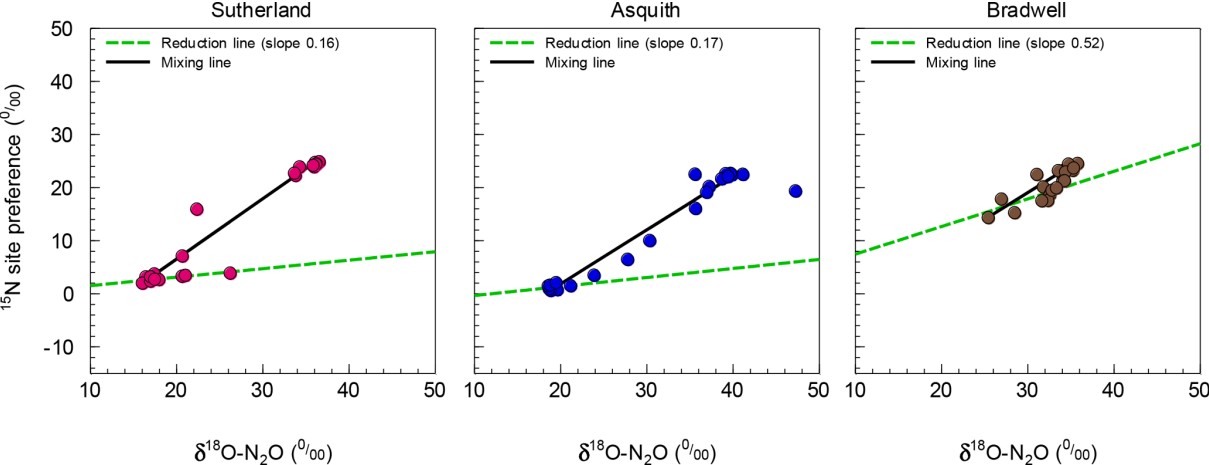

**Figure 3.** Isotopomer map to determine the source partitioning of $N_2O$ derived from nitrification versus denitrification using mean ($n = 4$) $^{15}N$ site preference (SP) and $\delta^{18}O$ of $N_2O$. The linear mixed model approach was based on Deppe et al. (2017) and Lewicka-Szczebak et al. (2017), but with mixing line endmembers and reduction line slopes derived from our data.

observable when conditions are favourable, as evidenced for the Bradwell soil. We echo earlier proposals made by Ostrom et al. (2007) and suggest that the current knowledge and understanding of $^{15}N_2O$ isotopomers may have inherent biases due to methodological focus on high-flux scenarios – where the rates of $N_2O$ reduction are minor and likely not of sufficient magnitude to alter isotopomer and SP data. Relatively few studies have focused on lower flux scenarios when the rates of $N_2O$ reduction relative to production may exert more of an influence on SP. Our findings support the hypothesis that $N_2O$ reduction is a minor process influencing SP during conditions of high soil $N_2O$ flux but may be more important for conditions with low $N_2O$ flux (Ostrom et al., 2007).

Due to the wide range of reduction slopes observed in our study – and the differences for how SP is influenced in conditions with high flux vs. low flux – we argue that using a single average reduction slope is insufficient for best predicting $N_2O$ reduction. This finding echoes earlier work which suggested that during soil conditions when processes of $N_2O$ production and reduction occur simultaneously, the reduction line approach may be limited (Decock and Six, 2013b). It is recommended that further research better quantify the conditions that promote $N_2O$ reduction for improved $N_2O$ source predictions. This could be particularly important for assessing microbial source pathways of $N_2O$ production and consumption across seasonal and spatial scales because sustained periods of low flux are not uncommon.

## 3.4 Source pathway partitioning and modelling

Using the pooled data from the isotopomer maps to predict source partitions, linear models were developed that fit the transitions for nitrification-derived $N_2O$ ($R^2 = 0.65$, $p<0.001$) and denitrification-derived $N_2O$ ($R^2 = 0.65$, $p<0.001$; Fig. 4) with coefficients of variation and root-

mean-square errors of 0.10 and 0.20, respectively. The models predict that over a soil moisture range of 53% to 78% WFPS, the source partitioning rapidly changed from nitrification- to denitrification-dominated $N_2O$ production. At soil moisture levels <53% WFPS, $N_2O$ was predominately attributed to nitrification ($F_N = 1$) but thereafter decreased rapidly, according to Eq. (6),

$$F_N = 3.19 - 0.041x, \tag{6}$$

until a WFPS of 78%. This result was mirrored by the increase in $N_2O$ attributed to denitrification at a WFPS of 53% according to Eq. (7),

$$F_D = -2.19 + 0.041x, \tag{7}$$

until $F_D = 1\%$ at 78% and higher WFPS. These relationships exemplify the sensitivity of $N_2O$ production pathways to soil moisture changes. For instance, during the transition, a change in soil moisture as little as 10% (i.e., from 55% to 65% WFPS) is predicted to lower nitrification-derived $N_2O$ production by 56% but increase denitrification-derived $N_2O$ by more than 7-fold (Fig. 4). Consequently, the linear models presented here may help other researchers estimate $N_2O$ source partitioning when soil moisture falls within the transition from nitrification to denitrification.

As a check, the soil-specific approach presented here was compared to the independent endmember or slope approach commonly used by other researchers (Deppe et al., 2017; Lewicka-Szczebak et al., 2017). Isotopomer maps were calculated using independent literature-derived values (see Supplemtn Fig. S1), with the endmembers set at −2.4 to 34.4 for $SP_D$ to $SP_N$ and 11.1 to 43.0 for $\delta^{18}O_D$ to $\delta^{18}O_N$ and a reduction slope of 0.33 (Lewicka-Szczebak et al., 2017). Using the literature-derived endmembers overestimated the contribution of denitrification-derived $N_2O$ under very dry

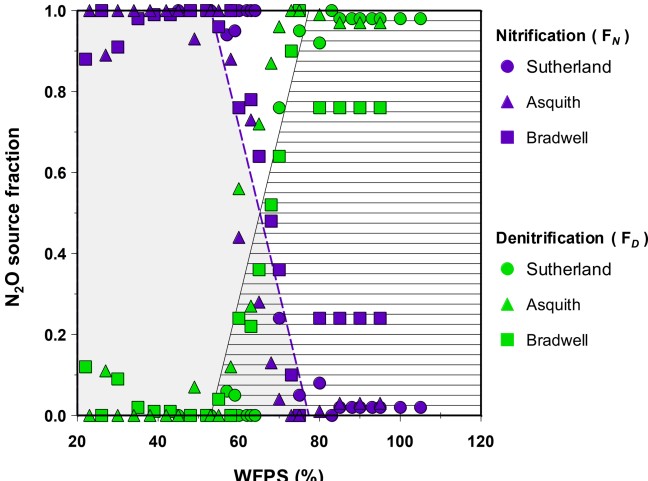

**Figure 4.** Fraction of emitted $N_2O$ that was attributed to nitrification ($F_N$; shaded grey area) or denitrification ($F_D$; lined area) based on the isotopomer mixing model (data points). Note: the dashed purple line denotes the predicted $F_N$ (Eq. 6); the solid black line denotes the predicted $F_D$ (Eq. 7).

soil conditions (i.e., 20 % to 40 % WFPS) – indicating that up to 40 % of $N_2O$ produced under these conditions was a result of denitrification (Fig. S1) – a contradiction to common knowledge (Butterbach-Bahl et al., 2013; Davidson et al. 1991). In our case, $N_2O$ source partitioning using soil-specific endmembers provided an advantage over using independent endmembers because the endmembers for the Bradwell soil were different from the literature-derived values, likely due to real soil biological processes such as microbial communities, the low rate of production, or soil heterogeneity (Decock and Six, 2013a; Lewicka-Szczebak et al., 2014). Nonetheless, we recommend that future research aims to develop more advanced models that take into account variability or more nuanced isotope effects.

Clearly, soil moisture change during the transition is a key regulator of which pathway dominantly produces $N_2O$ – be it nitrification, denitrification, or a mixture of both. Our results largely support the foundational studies that established the relationship between soil moisture and $N_2O$ emissions (Davidson, 1991; Linn and Doran, 1984); however, we provide a method that moves beyond just inferring $N_2O$ source pathways towards quantifying the pathway contributions over a range of soil moisture – and does so without having to add a [15]N label.

## 4   Conclusions

Determining the production pathways of soil-derived $N_2O$ is a worthwhile goal, as there is potential to manage soils in ways that lead to reduced nitrification or denitrification during periods of risk for $N_2O$ loss – thereby mitigating emissions of a potent greenhouse gas. We show that isotopomer

data have the potential to provide progress towards this goal. Measuring [15]$N_2O$ isotopomers enabled a more precise evaluation of the relationship between soil moisture and $N_2O$ production, and we present a source fraction model for key soil moisture ranges. In general, our results support earlier assumptions about the relationships between moisture and $N_2O$ production pathways but can help move beyond inferring towards quantifying relative source pathways. Clearly, soil moisture level during "the transition zone" is a key regulator of which pathway predominates – be it nitrification, denitrification, or a mixture of both. Hence, the models presented herein should be useful for other researchers in estimating contributions of nitrification versus denitrification when soil WFPS ranges from 53 % to 78 %.

One known caveat when using the isotopomer method for source pathway quantification is the isotope effect of $N_2O$ reduction. Previous researchers have attempted to address this limitation by using an average reduction slope and linear mixed model approach, but due to the wide range of reduction slopes observed in our study – and the differences for how denitrification SP is influenced in conditions with high $N_2O$ flux vs. low flux – we argue that using a single average reduction slope is insufficient for best predicting $N_2O$ reduction. It is recommended that further research better quantify the conditions which influence $N_2O$ reduction and its sensitivity on denitrification SP values for improved $N_2O$ source predictions. The creation of larger isotope databases would contribute to the development of more advanced models that take into account variability or more nuanced isotope effects.

**Data availability.** The data that support the findings of this study are available by request from the corresponding author (Kate Congreves).

**Supplement.** The supplement related to this article is available online at: https://doi.org/10.5194/soil-5-1-2019-supplement.

**Author contributions.** KAC and REF designed the experiment, and TP carried it out. KAC prepared the paper, with contributions from REF and TP.

**Competing interests.** The authors declare that they have no conflict of interest.

**Acknowledgements.** Richard E. Farrell is the co-director of the Prairie Environmental Agronomy Research Laboratory (PEARL) and director of the agricultural Greenhouse Gas Analysis Laboratory (agGAL), which provided analytical support for this project. The authors are grateful to Frank Krijnen and Darin Richman for technical help in the lab. Thank you to J. Diane Knight and Melissa M. Arcand for reviewing early drafts of the paper.

**Financial support.** This research has been supported by the University of Saskatchewan College of Agriculture and Bioresources via a Martin Agricultural Trust Fund award to Katelyn A. Congreves and Richard E. Farrell (grant no. 349252) and by the Natural Sciences and Engineering Research Council of Canada via a Discovery Grant award to Katelyn A. Congreves (grant no. RGPIN 2018-04953).

**Review statement.** This paper was edited by Steven Sleutel and reviewed by two anonymous referees.

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

**Remarks from the language copy-editor**

**Remarks from the typesetter**