# Peer review of "A new look at an old concept: Using 15N2O isotopomers to understand the relationship between soil moisture and N2O production pathways"

_SOIL, 2019_

## Referee Comment (RC1) · Anonymous Referee #1 · 11 Jun 2019

SOIL Discuss., https://doi.org/10.5194/soil-2019-27

Revisiting the relationship between soil moisture and N2O production pathways by measuring 15N2O isotopomers

General comments

The authors present a very nice and high quality dataset of N2O isotopomers from soil incubated over a gradient of moisture content. The study, however, has some major shortcomings in relating the dataset to the state of the art in N2O research. I encourage the authors to elaborate in 3 areas: 1) Latest approaches to interpret N2O isotopomer data 2) Consultation of literature on the effect of soil moisture on sources of N2O based

on isotope tracer work 3) Literature on factors controlling N2O reduction With a more in depth analysis of the data and discussion of the results in relation to current literature, I believe this study can become a valuable and much appreciated contribution to the discipline.

The reviewer has no intentions of promoting or favoring own or colleagues' work in the comments. The cited literature is intended as a resource and starting point for a more in-depth literature search.

Specific comments

Title:

P 1 The word 'revisiting' in the title implies to me that our understanding was wrong, but the isotopomers confirm what we already knew.

Abstract:

P 1 Line 14: the authors mention 'three soils'. I suggest adding a sentence explaining the difference between the three soils

P 1 Line 24: I assume x is soil moisture in this equation. Please specify and explain to the reader the potential relevance or importance of these equations

Introduction:

P 2 Lines 3-4 and Lines 8-13: There are several studies that investigated the effect of soil moisture on mechanisms underlying N2O emissions using 15N tracers. A few examples: - Stevens et al. 1997. Measuring the contributions of nitrification and denitrification to the flux of nitrous oxide from soil. Soil Biology and Biochemistry 29: 139-151 - Bateman and Baggs 2005. Contributions of nitrification and denitrification to N2O emissions from soils at different water-filled pore space. Biology and Fertility of Soils 41: 379-388

P 2 Line 19: Early studies on the use of isotopomers appeared in the early years 2000

by Ostrom et al., Well et al., and Toyoda et al. Please cite key early studies on the use of N2O isotopomers to source partition N2O.

Materials and methods: P 4 Line 25 – P 5 Line 20: A number of studies have been published on how to interpret N2O isotopomer data. Lewicka-Szczeback published an elegant method for calculating N2O from nitrification, denitrification and the fraction of N2O reduced to N2 based on SP and d18O of N2O. Details on the calculation approach can be found here. I recommend that the authors revise their calculation of the sources of N2O based on more recently published approaches. https://www.researchgate.net/publication/328135133_Mapping_approach_model_after_Lewicka-Szczebak_et_al_2017_-_detailed_description_of_calculation_procedures

The approach used by the authors has some major limitations, outlined below.

1) The authors use soil-specific end-members in their isotope mass balance, based on data from their experiment. While it cannot be excluded that isotope values characteristic of nitrification and denitrification are to some extent soil-dependent, the authors' approach relies on the assumption that at low moisture content, nitrification was the sole source of N2O, while denitrification was assumed to be the sole source of N2O at one of the medium range moisture contents. There is no independent measurement of the contribution of nitrification, denitrification and N2O reduction to N2. Limitations and assumptions of their approach need to be clearly stated.

Whether end-members are likely soil-dependent was discussed in a literature review by Decock and Six 2013. How reliable is the intramolecular distribution of 15N in N2O to source partition N2O emitted from soil? Soil Biology and Biochemistry 65: 114-127. Empirical studies since have further tested the effect of soil on end members, for example, Lewicka-Szczebak et al. 2014. Experimental determinations of isotopic fractionation factors associated with N2O production and reduction during denitrification in soils. Geochimica et Cosmochimica Acta 134:55–73. The results of the presented study should be discussed in relation to other studies published on this topic.

2) The authors use the relationship between d18O and SP at higher soil moisture content as the line representative of N2O reduction. It should be noted, however, that a N2O reduction line is only applicable if N2O reduction was the only process affecting N2O. In the presented experiment, N2O production and reduction likely occurred simultaneously. How simultaneous production and reduction of N2O affects isotope maps is discussed in great detail in Decock and Six. 2013. On the potential of d18O and d15N to assess N2O reduction to N2. European journal of soil science 64:610-620.

Results and discussion

P 6 Figure 2: It would be useful to see results of a statistical analysis on the effect of soil on N2O fluxes and isotopomer values across the moisture gradient.

In addition, it would be interesting to see the fraction of N2O derived from nitrification, denitrification and N2O reduction for each soil over the moisture gradient, including statistical analysis.

P 7 Line 16-18: I agree that a greater contribution of N2O reduction is a likely explanation for the observed results. The approach by Lewicka-Szczeback would allow the authors to calculate the fraction of N2O reduced to N2 based on the isotopomer data.

P 7 Line 22: A lot of literature has been published on factors controlling complete denitrification. See for example - Butterbach-Bahl et al. 2013. Nitrous oxide emissions from soil: How well do we understand the processes and their controls? Phil. Trans. R. Soc. B 2013 368, 20130122, - Groffman et al. 2006. Methods for measuring denitrification: Diverse approaches to a difficult problem. Ecological Applications 16:2091–2122 - and references therein.

P 7 Line 29-32: It is very likely that multiple processes underlying N2O emissions acted simultaneously to cause a higher than expected SP value. It needs to be very clear from the discussion that there was no independent measurement of nitrification, denitrification and N2O reduction to N2. To avoid this confounding factor in data interpretation, I strongly recommend the authors to use end-members from the literature for data-interpretation. Various studies have reviewed and summarized data for such end-members, for example: - Decock and Six 2013. How reliable is the intramolecular distribution of 15N in N2O to source partition N2O emitted from soil? Soil Biology and Biochemistry 65: 114-127; - Ostrom and Ostrom 2017. Mining the isotopic complexity of nitrous oxide: a review of challenges and opportunities. Biogeochemistry 132:359–372; - Denk et al. 2017. The nitrogen cycle: A review of isotope effects and isotope modeling approaches. Soil Biology and Biochemistry 105:121-137.

P8 Line 1 – P 9 Line 9: Here and elsewhere, please edit based on previous comments.

P 9 Line 10-27: This is an interesting analysis. I am interested to see models relating soil moisture to sources of N2O based on updated source calculations in line with the most recent literature. Based on the raw isotope data, I suspect a significant moisture by soil interaction with respect to sources of N2O. Statistical tests for such an interaction should be shown. Such an interaction may also have implications for the modeling approach in section 3.4 of this paper.

P 9 Line 27-28: Please refer to isotope tracer work, as suggested earlier.

Conclusion

Please edit commensurate with previous comments.

Technical corrections

None observed.

---

## Referee Comment (RC2) · Anonymous Referee #2 · 7 Jul 2019

Overall the data set is very interesting and will be of interest to readers.

The title uses the word 'revisiting'. While the introductory text notes the relationship between nitrification and denitrification processes with respect to soil moisture there is no prior evidence/studies introduced with respect to isotopomers and soil moisture. Thus the title may require suitable amendment or the introduction requires some additional information.

The introduction is nicely succinct and clear with respect to the problems associated with emissions of N2O, the role of soil moisture as a driver of N2O emissions, and the basics of isotopomers of N2O as linked to nitrification and denitrification. A reference

for terminology used would be good.

In the materials and methods sections: - were the soils sieved? I assume so seeing as they were placed in Petri dishes, thus what was the mesh size? - what was the randomised block design? There are 3 soils and 4 replicates but how many water treatments (WFPS treatments) and what were they? It appears looking at Fig 2 that there are about 16 WFPS treatments. -Note how N2O fluxes were determined. Assume it was just the one gas sample used, so there are I assume assumptions about linearity. - For the Piccaro CRDS is there a maximum/minimum N2O concentration? Were SP effects constant over a range of concentration?

The results and discussion are well considered and it is good the authors have considered N2O reduction effects on SP and interpreted results accordingly. For clarity, in the figure captions please state if the data presented are means or single points etc.

---

## Author Response (AR1)

***Response to Topical Editor***

Topical Editor Decision: Publish subject to minor revisions (review by editor) (30 Jul 2019) by Steven Sleutel

Comments to the Author:
The authors have very well accommodated referee requests. I agree with the proposed modifications to the manuscript. Only a few minor points of attention remain:

1) The abstract is too long: even detail is provided on specifics of the lab incubations. This part can be condensed without loss of information.

   **Authors:** we have condensed the abstract from 333 to 301 words; we removed the specifics of the soil incubations, and replaced with condensed text. See track changes for the revised parts on page 1/line 14-16.

2) P2 2nd line of the introduction: remove . after Environment Canada

   **Authors:** Done.

3) P2 L24 comma after 'Baggs'

   **Authors:** Done.

4) P3 L9-12 'The term "isotopomer" is used herein to indicate molecules of the same mass in which the trace isotopes are arranged differently. This differs from "isotopologue", which is a more general term referring to molecules that differ in isotopic composition (Ostrom and Ostrom, 2012).' Could be omitted.

   **Authors:** We have deleted the text from the Introduction, but moved the reference to Ostrom and Ostrom (2012) to Page 2/Line 20–21. In addition, we have added a second reference regarding terminology (McNaught and Wilkinson, 1997). Please note that these references were added to the text in response to a comment from Reviewer No. 2 (Comment #2).

5) It would indeed be a good idea to provide 'the new fig 5' as supplementary material and refer to it in the text. While your approach is plausible, like referee 1 most readers will be at first skeptic about setting SP endmembers based on these bulk measurements of soil emitted N2O. Fig. 5 clearly demonstrates that in fact using literature based SP endmembers yields unrealistic N2O source fraction calculations. This is an important point that does deserve a bit more attention in the discussion.

   **Authors:** We have added the figure to the manuscript as Supplemental Information and have included a short discussion of its importance to Section 3.4 of the Discussion:

   "As a check, the soil-specific approach presented here was compared to the independent endmember/slope approach commonly used by other researchers (Deppe et al., 2017; Lewicka-Szczebak et al., 2017). Isotopomer maps were calculated using independent literature-derived values (see Fig. S1) with the endmembers set at -2.4 to 34.4 for $SP_D$ to $SP_N$, and 11.1 to 43.0 for $\delta^{18}O_D$ to $\delta^{18}O_N$; and a reduction slope of 0.33 (Lewicka-Szczebak et al., 2017). Using the literature-derived endmembers overestimated the contribution of denitrification-derived N2O under very dry soil conditions (i.e., 20 to 40% WFPS)—indicating that up to 40% of N2O produced under these conditions was a result of denitrification (Fig. S1)—a contradiction to common knowledge (Butterbach-Bahl et al., 2013; Davidson et al. 1991). In our case, N2O source partitioning using soil-specific endmembers provided an advantage over using independent endmembers because the endmembers for the Bradwell soil were different from the literature-derived values; likely due to real soil biological processes such as microbial communities, the low rate of production, or soil heterogeneity (Decock and Six, 2013a; Lewicka-Szczebak et al., 2014). Nonetheless, we recommend that future research aims to develop more advanced models that take into account variability or more nuanced isotope effects." **(added at Page 10/Line 19)**

6) Fig. 4: Please clarify in the caption: are these observations from the three soils combined? I suspect so, but this is not immediately clear yet.

**Authors:** You are correct in your assessment. We have revised the figure and its caption to indicate this more clearly. **(Page 10 of the revised manuscript)**

7) The first line of the conclusions section seems a bit redundant to me – belongs rather in an abstract – but the authors can decide upon this.

**Authors:** Whereas we can see your point, we prefer to leave this in the Conclusions section. We just feel that this helps drive home why we think it is important to investigate the source partitioning of $N_2O$ emissions.

**Response to Reviewer Comments**

SOIL Discuss., https://doi.org/10.5194/soil-2019-27-RC1, 2019 © Author(s) 2019. This work is distributed under the Creative Commons Attribution 4.0 License.
Revisiting the relationship between soil moisture and N2O production pathways by measuring 15N2O isotopomers

**General comments**

The authors present a very nice and high quality dataset of N2O isotopomers from soil incubated over a gradient of moisture content. The study, however, has some major shortcomings in relating the dataset to the state of the art in N2O research. I encourage the authors to elaborate in 3 areas: 1) Latest approaches to interpret N2O isotopomer data 2) Consultation of literature on the effect of soil moisture on sources of N2O based on isotope tracer work 3) Literature on factors controlling N2O reduction With a more in depth analysis of the data and discussion of the results in relation to current literature, I believe this study can become a valuable and much appreciated contribution to the discipline.

Authors: Thank you for the comprehensive review. Revisions have been made to bolster references to previous literature, relating our work to previous $N_2O$ research.

The reviewer has no intentions of promoting or favoring own or colleagues' work in the comments. The cited literature is intended as a resource and starting point for a more in-depth literature search.

**Specific comments Title:**

1) P 1 The word 'revisiting' in the title implies to me that our understanding was wrong, but the isotopomers confirm what we already knew.

Authors: Both Reviewer#1 and #2 requested that the word 'revisiting be changed. We have given this some thought and can see the reviewers' point; thus we have changed the title to:

"A new look at an old concept: Using $^{15}N_2O$ isotopomers to understand the relationship between soil moisture and $N_2O$ production pathways".

**Abstract:**

2) P 1 Line 14: the authors mention 'three soils'. I suggest adding a sentence explaining the difference between the three soils

Authors: The following statement has been added to the revised text (page 1/line 15):

"For each of three soils—differing in nutrient levels, organic matter and texture—soil microcosms were arranged . . . "

3) P 1 Line 24: I assume x is soil moisture in this equation. Please specify and explain to the reader the potential relevance or importance of these equations

**Authors:** Revised to specify the equation variables for *x*, $F_N$, and $F_D$. As well, we added the following sentence to explain the potential relevance:

> "The presented equations may be helpful for other researchers to estimate $N_2O$ source partitioning when soil moisture falls within the transition from nitrification to denitrification". (added at page 1/line 25)

**Introduction:**

4) P 2 Lines 3-4 and Lines 8-13: There are several studies that investigated the effect of soil moisture on mechanisms underlying N2O emissions using 15N tracers. A few examples: - Stevens et al. 1997. Measuring the contributions of nitrification and denitrification to the flux of nitrous oxide from soil. Soil Biology and Biochemistry 29: 139-151 - Bateman and Baggs 2005. Contributions of nitrification and denitrification to N2O emissions from soils at different water-filled pore space. Biology and Fertility of Soils 41: 379-388

**Authors:** We have added a sentence acknowledging the use of [15]N tracers:

> "Indeed, our understanding of the relationship between $N_2O$ production and soil moisture has benefited greatly from the use of [15]N tracers (Bateman and Baggs, 2005; Stevens and Laughlin, 1997; Groffman et al., 2006)." **(added at page 2/line 1)**

However, the point we intended to make is that despite the advancements in understanding $N_2O$ and soil moisture, the *precise* relationships remains fairly unclear, especially during the transition. To better convey this message, the text was revised to include the following statements:

> "However, there remain surprising *grey-areas* in our understanding of the underlying mechanisms, one such area being the *precise* relationship between soil moisture and $N_2O$ production pathways, *especially during the transition from one dominant pathway to another* (Bateman and Baggs 2005). **(added at page 2/line 3)**

and

> "While previous research has provided important steps towards better quantifying the relationship using [15]N enrichment and acetylene inhibition techniques (Bateman and Baggs 2005), natural abundance [15]N techniques may provide superior information by imposing fewer confounding effects on step-wise N transformations." **(added at page 2/line 14)**

5) P 2 Line 19: Early studies on the use of isotopomers appeared in the early years 2000 by Ostrom et al., Well et al., and Toyoda et al. Please cite key early studies on the use of N2O isotopomers to source partition N2O.

**Authors:** We do refer to seminal papers by Toyoda, Ostrom, and Sutka further down this paragraph, but we have now revised the first sentence of the paragraph to acknowledge this body of early work up front.

> "….(Van Groenigen et al., 2015). Early work focused on the intramolecular distribution of [15]N within the linear $N_2O$ molecule (Sutka et al., 2006; Toyoda et al., 2005), investigations of atmospheric or oceanic $N_2O$ isotopomers (Popp et al., 2002; Toyoda and Yoshida, 1999; Yoshida and Toyoda, 2000), and soil emitted $N_2O$ istopomers (Perez et al., 2001; Yamulki et al., 2001)." **(added at page 2/line 22)**

6) **Materials and methods:** P 4 Line 25 – P 5 Line 20: A number of studies have been published on how to interpret N2O isotopomer data. Lewicka-Szczeback published an elegant method for calculating N2O from nitrification, denitrification and the fraction of N2O reduced to N2 based on SP and d18O of N2O. Details on the calculation approach can be found here. I recommend that the authors revise their calculation of the sources of N2O based on more recently published approaches. https://www.researchgate.net/publication/328135133_Mapping_approach_ model_after_Lewicka-Szczebak_et_al_2017_-_detailed_description_of_calculation_procedures

**Authors:** We confirmed that the calculation for the sources of $N_2O$ we performed were indeed the same approach as described by Lewicka-Szczebak. We cited the mixing model as described by Deppe et al. 2017, which is parallel to that used by Lewicka-Szczebak et al 2017. We now make reference to both papers that employed this approach. **(added at page 5/line 29)**

The mapping approach that we employed is that used by Deppe et al. (2017) and Lewicka-Szczebak et al. (2017) has been used before – but based on $\delta^{15}N$ SP and $\delta^{15}N$ bulk to estimate the fraction of bacterial $N_2O$ (Zou et al, 2014). As described by both Deppe et al (2017) and Lewicka-Szczebak et al (2017), they decided to base the mixing model on the relationship between $\delta^{15}N$ SP and $\delta^{18}O$ values (rather than bulk $^{15}N$) for more robust interpretations; accordingly, we followed their recommendations.

7) The approach used by the authors has some major limitations, outlined below.

The authors use soil-specific end-members in their isotope mass balance, based on data from their experiment. While it cannot be excluded that isotope values characteristic of nitrification and denitrification are to some extent soil-dependent, the authors' approach relies on the assumption that at low moisture content, nitrification was the sole source of N2O, while denitrification was assumed to be the sole source of N2O at one of the medium range moisture contents. There is no independent measurement of the contribution of nitrification, denitrification and N2O reduction to N2. Limitations and assumptions of their approach need to be clearly stated.

**Authors:** As requested, we added an explanation, and acknowledge the underlying assumptions and limitations (see page 5/lines 19–29 of the revised manuscript):

"Rather than relying on average literature-derived endmembers like previous work (Deppe et al., 2017; Lewicka-Szczebak et al., 2017), we used soil-specific endmembers derived from our data to perform the linear mixed model. This is because we measured a wide range of soil WFPS treatments with high frequency between dry and moist conditions for each soil, enabling us to determine the point at which the $\delta15N$ SP or $\delta18O$ values either dropped or increased as soil WFPS changed – as precisely as the data permitted. This approach is consistent with earlier recommendations that data is collected at high enough frequencies to capture gradual changes in isotope values as influenced by traditional proxies (i.e., gradual changes in soil WFPS) (Decock and Six, 2013a). However, it must be noted that the underlying assumption is that the soil-specific endmembers are more reflective of transition from nitrification to denitrification in each of the soils tested herein, than general literature-derived endmembers would be for any one soil. Moreover, it is assumed that the endmembers represent $N_2O$ fluxes when the sole source was either nitrification or denitrification." **(added at page 5/line 12)**

8) Whether end-members are likely soil-dependent was discussed in a literature review by Decock and Six 2013. How reliable is the intramolecular distribution of 15N in N2O to source partition N2O emitted from soil? Soil Biology and Biochemistry 65: 114- 127. Empirical studies since have further tested the effect of soil on end members, for example, Lewicka-Szczebak et al. 2014. Experimental determinations of isotopic fractionation factors associated with N2O production and reduction during denitrification in soils. Geochimica et Cosmochimica Acta 134:55–73. The results of the presented study should be discussed in relation to other studies published on this topic.

**Authors:** Revisions have been made to discuss this study in relation to others. We reference the review by Decock and Six (2013a and b), and Lewicka et al (2014):

"Despite similarities among soils in the robust patterns of how SP values are influenced by soil moisture (Fig. 2; Table 2), SP exhibited a significant ($P < 0.0001$) soil $\times$ moisture region interaction. This finding agrees with earlier suggestions that, at finer scales, the $^{15}N_2O$ isotopic signatures and SP values are likely regulated by the active soil microbial community, process rates, soil heterogeneity (Decock and Six, 2013a; Lewicka-Szczebak et al., 2014)." **(added at page 8/line 17)**

9) The authors use the relationship between d18O and SP at higher soil moisture content as the line representative of N2O reduction. It should be noted, however, that a N2O reduction line is only applicable if N2O reduction was the

only process affecting N2O. In the presented experiment, N2O production and reduction likely occurred simultaneously. How simultaneous production and reduction of N2O affects isotope maps is discussed in great detail in Decock and Six. 2013. On the potential of d18O and d15N to assess N2O reduction to N2. European journal of soil science 64:610-620.

**Authors:** We have revised the text to address the reviewer's concerns; transparently identify the limitations of our study; and discuss our study in context to others:

"Previously, the fractionation of SP due to $N_2O$ reduction was constrained to a variation of -2‰ to -8‰ (Jinuntuya-Nortman et al., 2008; Lewicka-Szczebak et al., 2014; Well and Flessa, 2009). Ostrom et al. (2007) showed that … " **(added at page 8/line 26)**

"Reduction slopes for our three soils averaged 0.28, which is similar to the literature-derived average of 0.35 or. 0.33 used by Deppe et al. (2017) and Lewicka-Szczebak et al. (2014), respectively . . . " **(added at page 8/line 34)**

"This finding echoes earlier work which suggested that during soil conditions when processes of $N_2O$ production and reduction occur simultaneously, the reduction line approach may be limited (Decock and Six, 2013b)." **(added at page 9/line 17)**

**Results and discussion**

10) P 6 Figure 2: It would be useful to see results of a statistical analysis on the effect of soil on N2O fluxes and isotopomer values across the moisture gradient. In addition, it would be interesting to see the fraction of N2O derived from nitrification, denitrification and N2O reduction for each soil over the moisture gradient, including statistical analysis.

**Authors:** First, we apologize for a miscommunication regarding $N_2O$ fluxes; i.e., we did not measure $N_2O$ fluxes; rather, we measured the total amount of $N_2O$ produced during the 24-h incubation. This has been made more clear in the revised text (Section 3.1; see response to comment #4 by Reviewer #2).

As for the Reviewer's second point, we are not entirely sure what is being asked for here. To evaluate the influence of soil moisture on $N_2O$ production and isotopomers, we analyzed the relationship between WFPS and SP or source fraction during key moisture gradients via linear regressions – see Figure 4 and Table 2. As such, the statistical analysis was approached in two ways: 1) linear regression to characterize $N_2O$ isotopic changes as influenced by moisture (see Table 2), and 2) by developing linear models within the transition zone to model the changes in $N_2O$ source fraction (Eq 6 & 7), see Fig 4. We believe the way we approached the analyses is best suited for our objectives of this study. To evaluate the influence of soil attributes on $N_2O$ and isotopomers, a future study could be designed and carried out to best test numerous different soil types and soil attributes on $N_2O$ and isotopomers.

11) P 7 Line 16-18: I agree that a greater contribution of N2O reduction is a likely explanation for the observed results. The approach by Lewicka-Szczeback would allow the authors to calculate the fraction of N2O reduced to N2 based on the isotopomer data.

**Authors:** Thanks for the suggestion. We have adopted this approach and revised the text to reflect the outcome of the analysis:

"Correspondingly, using the mapping model approach, we estimated much larger fractions of $N_2O$ were reduced to $N_2$ at 95% WFPS in the Bradwell soil (0.47), compared to the Sutherland or Asquith soils (0.13 to 0.14)." **(added at page 7/line 34)**

12) P 7 Line 22: A lot of literature has been published on factors controlling complete denitrification. See for example - Butterbach-Bahl et al. 2013. Nitrous oxide emissions from soil: How well do we understand the processes and their controls? Phil. Trans. R. Soc. B 2013 368, 20130122, - Groffman et al. 2006. Methods for measuring denitrification: Diverse approaches to a difficult problem. Ecological Applications 16:2091–2122 - and references

therein.

13) P 7 Line 29-32: It is very likely that multiple processes underlying N2O emissions acted simultaneously to cause a higher than expected SP value. It needs to be very clear from the discussion that there was no independent measurement of nitrification, denitrification and N2O reduction to N2. To avoid this confounding factor in data interpretation, I strongly recommend the authors to use end-members from the literature for data-interpretation. Various studies have reviewed and summarized data for such end-members, for example: - Decock and Six 2013. How reliable is the intramolecular distribution of 15N in N2O to source partition N2O emitted from soil? Soil Biology and Biochemistry 65: 114-127; - Ostrom and Ostrom 2017. Mining the isotopic complexity of nitrous oxide: a review of challenges and opportunities. Biogeochemistry 132:359– 372; - Denk et al. 2017. The nitrogen cycle: A review of isotope effects and isotope modeling approaches. Soil Biology and Biochemistry 105:121-137.

**Authors:** The assumptions, explanations, and limitations are now more clearly described (see authors' response to Comment #7, above).

To check if data interpretation would be influenced by using the soil-specific approach *vs* the independent endmember/slope approach, isotopomer maps were also calculated using independent literature-derived values (see below Fig). Literature-derived endmembers were set at -2.4 to 34.4 for $SP_D$ to $SP_N$, and 11.1 to 43.0 for $\delta^{18}O_D$ to $\delta^{18}O_N$ ; a reduction slope of 0.33 (Lewicka-Szczebak et al., 2017). For our study, using literature-derived endmembers was deemed inappropriate because it overestimated the contribution of denitrification-derived N2O under very dry soil conditions (i.e., 20 to 40% WFPS), to fractions of up to 40% of N2O produced (see figure below) – a result which is in contradiction to common knowledge (Davidson et al. 1991). In our case, N2O source partitioning using soil-specific endmembers provided an advantage over using independent endmembers because certain endmembers (Bradwell soil) were different from expected/literature-derived values; likely due to real soil biological processes such as microbial communities, the low rate of production, or soil heterogeneity (Decock and Six, 2013a; Lewicka-Szczebak et al., 2014). So, it makes more sense in our case to use soil-specific endmembers (especially because we have the isotopic data over a fine-scale WFPS change, enabling us to pin-point soil-specific endmembers as best we can). We are transparent about the approach, the assumptions, and limitations (see revisions to page 5 which clearly explains that the endmembers are assumed to represent sole nitrification or denitrification). We are open to including the below figure and explanation as a Supplemental material if you deem it necessary, but please note that our objectives were not to compare different endmember approaches. Rather, our goal was to use a reasonable endmember approach to estimate source partitions, and evaluate how soil moisture affects N2O production. See attached Figure.

14) P8 Line 1 – P 9 Line 9: Here and elsewhere, please edit based on previous comments.

**Authors:** This section has been revised to reflect our response to the Reviewer's previous comments. In brief, these revisions have to do with placing our results in better context with the published literature. (See tracked changes in accompanying revised manuscript.)

15) P 9 Line 10-27: This is an interesting analysis. I am interested to see models relating soil moisture to sources of N2O based on updated source calculations in line with the most recent literature. Based on the raw isotope data, I suspect a significant moisture by soil interaction with respect to sources of N2O. Statistical tests for such an interaction should be shown. Such an interaction may also have implications for the modeling approach in section 3.4 of this paper.

**Authors:** See response to Comment #8 above in which we added a statement showing indicating that site preference exhibited a significant ($P < 0.0001$) soil × moisture region interaction. This provides additional support for our decision to use site-specific endmembers to calculate the source fractions attributable to nitrification and denitrification in our modeling. Further, the site-specific end-member approach effectively normalizes the source partitions among soils, enabling data to be pooled for the modeling approach (Fig 4). Refer to discussion under Comment 13.

16) P 9 Line 27-28: Please refer to isotope tracer work, as suggested earlier.

**Authors:** The text was revised to reference isotope tracer work in response to Comment # 4 above; however, we do not feel that it needs to be repeated here. Our purpose was to determine whether we could use $^{15}$N isotopomers—i.e., natural abundance, not $^{15}$N tracers—to better elucidate the relationship between soil moisture and N$_2$O production pathways. Nevertheless, we have revised the text in this section to read:

"Our results largely support the foundational studies that established the relationship between soil moisture and N$_2$O emissions (Davidson, 1991; Linn and Doran, 1984); however, we provide a method that moves beyond just inferring N$_2$O source pathways towards quantifying the pathway contributions over a range of soil moisture—and does so without having to add a $^{15}$N label." **(underlined text added at page 11/line 10)**

17) **Conclusion**
Please edit commensurate with previous comments.

**Authors:** The conclusions have been revised to reflect previous comments.

18) **Technical corrections**
None observed.

**Anonymous Referee #2**

Overall the data set is very interesting and will be of interest to readers.

1) The title uses the word 'revisiting'. While the introductory text notes the relationship be- tween nitrification and denitrification processes with respect to soil moisture there is no prior evidence/studies introduced with respect to isotopomers and soil moisture. Thus the title may require suitable amendment or the introduction requires some additional information.

**Authors:** Text has been added to the introduction to make reference to earlier studies that have evaluated the relationship between moisture and N$_2$O, and which have used $^{15}$N tracer techniques. No previous studies that we are aware of have used isotopomers to look at this relationship. We have modified the title to better convey our meaning:

"A new look at an old concept: Using $^{15}$N$_2$O isotopomers to understand the relationship between soil moisture and N$_2$O production pathways".

2) The introduction is nicely succinct and clear with respect to the problems associated with emissions of N2O, the role of soil moisture as a driver of N2O emissions, and the basics of isotopomers of N2O as linked to nitrification and denitrification. A reference for terminology used would be good.

**Authors:** we added a citation after the terminology reference, see page 3/line 8 (Ostrom and Ostrom 2012).

3) In the materials and methods sections: - were the soils sieved? I assume so seeing as they were placed in Petri dishes, thus what was the mesh size? - what was the randomised block design? There are 3 soils and 4 replicates but how many water treatments (WFPS treatments) and what were they? It appears looking at Fig 2 that there are about 16 WFPS treatments

**Authors:** yes the soils were sieved using a 2-mm mesh screen; this information has been added to the revised text (page 4/line 5). The final moisture range (40 to 105% WFPS) was based on data collected during a preliminary test in which soil moisture was varied from 10% to 105%. A lack of N$_2$O production when WFPS was <40% led us to limit the moisture range for the final study. The number of moisture treatments within the overall range varied depending on soil texture (21 for the Sutherland soil; 17 for the Asquith soil; and 16 for the Bradwell soil). We can add this information to the text, or add a supplemental table that lists the individual moisture treatments

for each soil (i.e., gravimetric soil water content and WFPS).

The soil microcosms were arranged using a completely randomized design (CRD) with four replicates. This information was **added at page 3/line 32**.

4) Note how N2O fluxes were determined. Assume it was just the one gas sample used, so there are I assume assumptions about linearity.

**Authors:** We apologize for the miscommunication. We did not measure $N_2O$ fluxes; rather, we measured the total amount of $N_2O$ produced during the 24-h incubation. The text has been revised (Section 3.1) to reflect this fact. We have changed the y-axis title in Figure 2 from "ng g$^{-1}$ d$^{-1}$" to "ng g$^{-1}$ 24-h$^{-1}$"; and have changed "$N_2O$ flux" to "$N_2O$ production" throughout the manuscript.

5) For the Picarro CRDS is there a maximum/minimum N2O concentration? Were SP effects constant over a range of concentration?

**Authors:** The minimum detectable $N_2O$ concentration of the G5131-*i* (*ca.* 0.05 ppb) is well below any of the concentrations we measured during the study (the minimum concentrations being at ambient levels). The maximum concentration we can measure without difficulty is about 1000 ppb; consequently, any sample determined to exceed this concentration (based on the GC analyses) was diluted with ultra-pure zero-air to a concentration of approximately 300 ppb for $^{15}N_2O$ isotopomer analysis. As well, during earlier testing of the instrument performance it was determined that site preference was independent of gas concentration. (Note: instrument performance is being described in a separate manuscript that is still in the works.)

6) The results and discussion are well considered and it is good the authors have considered N2O reduction effects on SP and interpreted results accordingly. For clarity, in the figure captions please state if the data presented are means or single points etc.

**Authors:** The data presented in Fig 2 are the means (n = 4) means and bars represent the standard error of the mean. This information has been added to the figure caption (**page 6**). Likewise, the data presented in Fig. 3 are the mean site preference values; this information has been added to the figure caption (**page 9**).

[revised manuscript text omitted]